# DECENTRALIZED DEEP LEARNING WITH ARBITRARY COMMUNICATION COMPRESSION

**Anastasia Koloskova**[*]
anastasia.koloskova@epfl.ch

**Tao Lin**[*]
tao.lin@epfl.ch

**Sebastian U. Stich**
sebastian.stich@epfl.ch

**Martin Jaggi**
martin.jaggi@epfl.ch

**EPFL**
Lausanne, Switzerland

## ABSTRACT

Decentralized training of deep learning models is a key element for enabling data privacy and on-device learning over networks, as well as for efficient scaling to large compute clusters. As current approaches are limited by network bandwidth, we propose the use of communication compression in the decentralized training context. We show that CHOCO-SGD achieves linear speedup in the number of workers for arbitrary high compression ratios on general *non-convex* functions, and non-IID training data. We demonstrate the practical performance of the algorithm in two key scenarios: the training of deep learning models (i) over decentralized user devices, connected by a peer-to-peer network and (ii) in a datacenter.

## 1 INTRODUCTION

Distributed machine learning—i.e. the training of machine learning models using distributed optimization algorithms—has recently enabled many successful applications in research and industry. Such methods offer two of the key success factors: 1) *computational scalability* by leveraging the simultaneous computational power of many devices, and 2) *data-locality*, the ability to perform joint training while keeping each part of the training data local to each participating device. Recent theoretical results indicate that decentralized schemes can be as efficient as the centralized approaches, at least when considering convergence of training loss vs. iterations (Scaman et al., 2017; 2018; Lian et al., 2017; Tang et al., 2018; Koloskova et al., 2019; Assran et al., 2019).

Gradient compression techniques have been proposed for the standard distributed training case (Alistarh et al., 2017; Wen et al., 2017; Lin et al., 2018; Wangni et al., 2018; Stich et al., 2018), to reduce the amount of data that has to be sent over each communication link in the network. For decentralized training of deep neural networks, Tang et al. (2018) introduce two algorithms (DCD, ECD) which allow for communication compression. However, both these algorithms are restrictive with respect to the used compression operators, only allowing for unbiased compressors and—more significantly—so far not supporting arbitrarily high compression ratios. We here study CHOCO-SGD—recently introduced for convex problems only (Koloskova et al., 2019)—which overcomes these constraints.

For the evaluation of our algorithm we in particular focus on the generalization performance (on the test-set) on standard machine learning benchmarks, hereby departing from previous work such as e.g. (Tang et al., 2018; Wang et al., 2019; Tang et al., 2019; Reisizadeh et al., 2019) that mostly considered training performance (on the train-set). We study two different scenarios: firstly, (i) training on a challenging peer-to-peer setting, where the training data is distributed over the training devices (and not allowed to move), similar to the federated learning setting (McMahan et al., 2017). We are again able to show speed-ups for CHOCO-SGD over the decentralized baseline (Lian et al., 2017) with much less communication overhead. Secondly, (ii) training in a datacenter setting, where decentralized communication patterns allow better scalability than centralized approaches. For this setting

---

[*]Equal contribution.

we show that communication efficient CHOCO-SGD can improve time-to-accuracy on large tasks, such as e.g. ImageNet training. However, when investigating the scaling of decentralized algorithms to larger number of nodes we observe that (all) decentralized schemes encounter difficulties and often do not reach the same (test and train) performance as centralized schemes. As these findings point out some deficiencies of current decentralized training schemes (and are not particular to our scheme) we think that reporting these results is a helpful contribution to the community to spur further research on decentralized training schemes that scale to large number of peers.

**Contributions.**   Our contributions can be summarized as:
- On the theory side, we are the first to show that CHOCO-SGD converges at rate $\mathcal{O}\left(1/\sqrt{nT} + 1/(\rho^2\delta T)^{2/3}\right)$ on non-convex smooth functions, where $n$ denotes the number of nodes, $T$ the number of iterations, $\rho$ the spectral gap of the mixing matrix and $\delta$ the compression ratio. The main term, $\mathcal{O}\left(1/\sqrt{nT}\right)$, matches with the centralized baselines with exact communication and shows a linear speedup in the number of workers $n$. Both $\rho$ and $\delta$ only affect the asymptotically smaller second term.
- On the practical side, we present a version of CHOCO-SGD with momentum and analyze its practical performance on two relevant scenarios:
  - for *on-device training* over a realistic peer-to-peer social network, where lowering the bandwidth requirements of joint training is especially impactful
  - in a datacenter setting for *computational scalability* of training deep learning models for resource efficiency and improved time-to-accuracy
- Lastly, we systematically investigate performance of the decentralized schemes when scaling to larger number of nodes and we point out some (shared) difficulties encountered by current decentralized learning approaches.

## 2  RELATED WORK

For the training in communication restricted settings a variety of methods have been proposed. For instance, decentralized schemes (Lian et al., 2017; Nedić et al., 2018; Koloskova et al., 2019), gradient compression (Seide et al., 2014; Strom, 2015; Alistarh et al., 2017; Wen et al., 2017; Lin et al., 2018; Wangni et al., 2018; Bernstein et al., 2018; Lin et al., 2018; Alistarh et al., 2018; Stich et al., 2018; Karimireddy et al., 2019), asynchronous methods (Recht et al., 2011; Assran et al., 2019) or performing multiple local SGD steps before averaging (Zhang et al., 2016; McMahan et al., 2017; Lin et al., 2020). This especially covers learning over decentralized data, as extensively studied in the federated learning literature for the centralized algorithms (McMahan et al., 2016). In this paper we advocate for combining decentralized SGD schemes with gradient compression.

**Decentralized SGD.** We in particular focus on approaches based on gossip averaging (Kempe et al., 2003; Xiao & Boyd, 2004; Boyd et al., 2006) whose convergence rate typically depends on the spectral gap $\rho \geq 0$ of the mixing matrix (Xiao & Boyd, 2004). Lian et al. (2017) combine SGD with gossip averaging and show that the leading term in the convergence rate $\mathcal{O}\left(1/\sqrt{nT}\right)$ is consistent with the convergence of the centralized mini-batch SGD (Dekel et al., 2012) and the spectral gap only affects the asymptotically smaller terms. Similar results have been observed very recently for related schemes (Scaman et al., 2017; 2018; Koloskova et al., 2019; Yu et al., 2019).

**Quantization.** Communication compression with quantization has been popularized in the deep learning community by the reported successes in (Seide et al., 2014; Strom, 2015). Theoretical guarantees were first established for schemes with unbiased compression (Alistarh et al., 2017; Wen et al., 2017; Wangni et al., 2018) but soon extended to biased compression (Bernstein et al., 2018) as well. Schemes with error correction work often best in practice and give the best theoretical gurantees (Lin et al., 2018; Alistarh et al., 2018; Stich et al., 2018; Karimireddy et al., 2019). Recently, also proximal updates and variance reduction have been studied in combination with quantized updates (Mishchenko et al., 2019; Horváth et al., 2019).

**Decentralized Optimization with Quantization.** It has been observed that gossip averaging can diverge (or not converge to the correct solution) in the presence of quantization noise (Xiao et al., 2005; Carli et al., 2007; Nedić et al., 2008; Dimakis et al., 2010; Carli et al., 2010b; Yuan et al., 2012). Reisizadeh et al. (2018) propose an algorithm that can still converge, though at a slower rate than the exact scheme. Another line of work proposed adaptive schemes (with increasing compression accuracy) that converge at the expense of higher communication cost (Carli et al., 2010a; Doan et al.,

2018; Berahas et al., 2019). For deep learning applications, Tang et al. (2018) proposed the DCD and ECD algorithms that converge at the same rate as the centralized baseline though only for *constant* compression ratio. The CHOCO-SGD algorithm that we consider in this work can deal with *arbitrary* high compression, and has been introduced in (Koloskova et al., 2019) but only been analyzed for convex functions. For non-convex functions we show a rate of $\mathcal{O}\big(1/\sqrt{nT} + 1/(\rho^2 \delta T)^{\frac{2}{3}}\big)$, where $\delta > 0$ measures the compression quality. Simultaneous work of Tang et al. (2019) introduced DeepSqueeze, an alternative method which also converges with arbitrary compression ratio. In our experiments, under the same amount of tuning, CHOCO-SGD achieves higher test accuracy.

## 3 CHOCO-SGD

In this section we formally introduce the decentralized optimization problem, compression operators, and the gossip-based stochastic optimization algorithm CHOCO-SGD from (Koloskova et al., 2019).

**Distributed Setup.** We consider optimization problems distributed across $n$ nodes of the form

$$f^\star := \min_{\mathbf{x} \in \mathbb{R}^d} \left[ f(\mathbf{x}) := \frac{1}{n} \sum_{i=1}^{n} f_i(\mathbf{x}) \right], \qquad f_i(\mathbf{x}) := \mathbb{E}_{\xi_i \sim D_i} F_i(\mathbf{x}, \xi_i), \qquad \forall i \in [n], \quad (1)$$

where $D_1, \ldots D_n$ are local distributions for sampling data which can be different on every node, $F_i \colon \mathbb{R}^d \times \Omega \to \mathbb{R}$ are possibly non-convex (and non-identical) loss functions. This setting covers the important case of empirical risk minimization in distributed machine learning and deep learning applications.

**Communication.** Every device is only allowed to communicate with its local neighbours defined by the network *topology*, given as a weighted graph $G = ([n], E)$, with edges $E$ representing the communication links along which messages (e.g. model updates) can be exchanged. We assign a positive weight $w_{ij}$ to every edge ($w_{ij} = 0$ for disconnected nodes $\{i, j\} \notin E$).

**Assumption 1** (Mixing matrix). *We assume that $W \in [0,1]^{n \times n}$, $(W)_{ij} = w_{ij}$ is a symmetric ($W = W^\top$) doubly stochastic ($W\mathbf{1} = \mathbf{1}, \mathbf{1}^\top W = \mathbf{1}^\top$) matrix with eigenvalues $1 = |\lambda_1(W)| > |\lambda_2(W)| \geq \cdots \geq |\lambda_n(W)|$ and spectral gap $\rho := 1 - |\lambda_2(W)| \in (0,1]$.*

In our experiments we set the weights based on the local node degrees: $w_{ij} = \max\{\deg(i), \deg(j)\}^{-1}$ for $\{i, j\} \in E$. This will not only guarantee $\rho > 0$ but these weights can easily be computed in a local fashion on each node (Xiao & Boyd, 2004).

**Compression.** We aim to only transmit *compressed* (e.g. quantized or sparsified) messages. We formalized this through the notion of compression operators that was e.g. also used in (Tang et al., 2018; Stich et al., 2018).

**Definition 3.1** (Compression operator). *$Q \colon \mathbb{R}^d \to \mathbb{R}^d$ is a compression operator if it satisfies*

$$\mathbb{E}_Q \|Q(\mathbf{x}) - \mathbf{x}\|^2 \leq (1 - \delta) \|\mathbf{x}\|^2, \qquad \forall \mathbf{x} \in \mathbb{R}^d, \quad (2)$$

*for a parameter $\delta > 0$. Here $\mathbb{E}_Q$ denotes the expectation over the internal randomness of operator $Q$.*

In contrast to the quantization operators used in e.g. (Alistarh et al., 2017; Horváth et al., 2019), compression operators defined as in (2) are not required to be unbiased and therefore supports a larger class of compression operators. Some examples can be found in (Koloskova et al., 2019) and we further discuss specific compression schemes in Section 5.

**Algorithm.** CHOCO-SGD is summarized in Algorithm 1. Every worker $i$ stores its own private variable $\mathbf{x}_i \in \mathbb{R}^d$ that is updated by a stochastic gradient step in part ② and a modified gossip averaging step on line 2. This step is a key element of the algorithm as it preserves the averages of the iterates even in presence of quantization noise (the compression errors are not discarded, but aggregated in the local variables $\mathbf{x}_i$, see also (Koloskova et al., 2019)). The nodes communicate with their neighbors in part ① and update the variables $\hat{\mathbf{x}}_j \in \mathbb{R}^d$ for all their neighbors $\{i, j\} \in E$ only using compressed updates. These $\hat{\mathbf{x}}_i$ are available to all the neighbours of the node $i$ and represent the 'publicly available' copies of the private $\mathbf{x}_i$, in general $\mathbf{x}_i \neq \hat{\mathbf{x}}_i$, due to the communication restrictions.

From an implementation aspect, it is worth highlighting that the communication part ① and the gradient computation part ② can both be executed in parallel because they are independent. Moreover,

---

**Algorithm 1** CHOCO-SGD (Koloskova et al., 2019)

---

**input:** Initial value $\overline{\mathbf{x}}^{(-\frac{1}{2})} \in \mathbb{R}^d$, $\mathbf{x}_i^{(-\frac{1}{2})} = \overline{\mathbf{x}}^{(-\frac{1}{2})}$ on each node $i \in [n]$, consensus stepsize $\gamma$, SGD stepsize $\eta$,
    communication graph $G = ([n], E)$ and mixing matrix $W$, initialize $\hat{\mathbf{x}}_i^{(0)} := \mathbf{0} \; \forall i \in [n]$

  1: **for** $t$ **in** $0 \ldots T-1$ **do** {*in parallel for all workers $i \in [n]$*}

  2:     $\mathbf{x}_i^{(t)} := \mathbf{x}_i^{(t-\frac{1}{2})} + \gamma \sum_{j:\{i,j\}\in E} w_{ij}\big(\hat{\mathbf{x}}_j^{(t)} - \hat{\mathbf{x}}_i^{(t)}\big)$              ◁ modified gossip averaging

      ⎧ 3:     $\mathbf{q}_i^{(t)} := Q(\mathbf{x}_i^{(t)} - \hat{\mathbf{x}}_i^{(t)})$                             ◁ compression

      ⎪ 4:     **for** neighbors $j\colon \{i,j\} \in E$ (including $\{i\} \in E$) **do**

① ⎨ 5:       Send $\mathbf{q}_i^{(t)}$ and receive $\mathbf{q}_j^{(t)}$                    ◁ communication

      ⎪ 6:       $\hat{\mathbf{x}}_j^{(t+1)} := \mathbf{q}_j^{(t)} + \hat{\mathbf{x}}_j^{(t)}$                      ◁ local update

      ⎩ 7:     **end for**

      ⎧ 8:     Sample $\xi_i^{(t)}$, compute gradient $\mathbf{g}_i^{(t)} := \nabla F_i(\mathbf{x}_i^{(t)}, \xi_i^{(t)})$

② ⎨ 9:     $\mathbf{x}_i^{(t+\frac{1}{2})} := \mathbf{x}_i^{(t)} - \eta \mathbf{g}_i^{(t)}$                       ◁ stochastic gradient update

  10: **end for**

---

each node only needs to store 3 vectors at most, independent of the number of neighbors (this might not be obvious from the notation used here for additinal clarity, for further details c.f. (Koloskova et al., 2019)). We further propose a momentum-version of CHOCO-SGD in Algorithm 2 (see also Section D for further details).

## 4   CONVERGENCE OF CHOCO-SGD ON SMOOTH NON-CONVEX PROBLEMS

As the first main contribution, we here extend the analysis of CHOCO-SGD to non-convex problems. For this we make the following technical assumptions:

**Assumption 2.** *Each function $f_i \colon \mathbb{R}^d \to \mathbb{R}$ for $i \in [n]$ is L-smooth, that is*

$$\|\nabla f_i(\mathbf{y}) - \nabla f_i(\mathbf{x})\| \le L \|\mathbf{y} - \mathbf{x}\| , \qquad \qquad \forall \mathbf{x}, \mathbf{y} \in \mathbb{R}^d, i \in [n],$$

*and the variance of the stochastic gradients is bounded on each worker:*

$$\mathbb{E}_{\xi_i} \|\nabla F_i(\mathbf{x}, \xi_i) - \nabla f_i(\mathbf{x})\|^2 \le \sigma_i^2 , \qquad \mathbb{E}_{\xi_i} \|\nabla F_i(\mathbf{x}, \xi_i)\|^2 \le G^2 , \qquad \forall \mathbf{x} \in \mathbb{R}^d, i \in [n], \qquad (3)$$

*where $\mathbb{E}_{\xi_i}[\cdot]$ denotes the expectation over $\xi_i \sim \mathcal{D}_i$. We also denote $\overline{\sigma}^2 := \frac{1}{n}\sum_{i=1}^n \sigma_i^2$ for convenience.*

**Theorem 4.1.** *Under Assumptions 1–2 there exists a constant stepsize $\eta$ and the consensus stepsize from (Koloskova et al., 2019), $\gamma := \frac{\rho^2 \delta}{16\rho + \rho^2 + 4\beta^2 + 2\rho\beta^2 - 8\rho\delta}$ with $\beta = \|I - W\|_2 \in [0, 2]$, such that the averaged iterates $\overline{\mathbf{x}}^{(t)} := \frac{1}{n}\sum_{i=1}^n \mathbf{x}_i^{(t)}$ of Algorithm 1 satisfy:*

$$\frac{1}{T+1}\sum_{t=0}^T \left\|\nabla f\big(\overline{\mathbf{x}}^{(t)}\big)\right\|_2^2 = \mathcal{O}\left( \left(\frac{LF_0 \overline{\sigma}^2}{n(T+1)}\right)^{1/2} + \left(\frac{GLF_0}{c(T+1)}\right)^{2/3} + \frac{LF_0}{T+1} \right)$$

*where $c := \frac{\rho^2 \delta}{82}$ denotes the convergence rate of the underlying consensus averaging scheme of (Koloskova et al., 2019), $F_0 := f(\overline{\mathbf{x}}^{(0)}) - f^\star$.*

This result shows that CHOCO-SGD converges as $\mathcal{O}\big(1/\sqrt{nT} + 1/(\rho^2\delta T)^{2/3}\big)$. The first term shows a linear speed-up compared to SGD on a single node, while compression and graph topology affect only the higher order second term. By differently choosing the stepsize $\eta := \sqrt{n}/\sqrt{T+1}$ we can recover asymptotic convergence rate of $\mathcal{O}\big(1/\sqrt{nT} + n/(\rho^4\delta^2 T)\big)$. For the proofs and convergence of the individual iterates $\mathbf{x}_i$ we refer to Appendix A.

## 5   COMPARISON TO BASELINES FOR VARIOUS COMPRESSION SCHEMES

In this section we experimentally compare CHOCO-SGD to the relevant baselines for a selection of commonly used compression operators. For the experiments we further leverage momentum in all implemented algorithms. The newly developed momentum version of CHOCO-SGD is given as Algorithm 2.

---

**Algorithm 2** CHOCO-SGD with Momentum

---

**input:** Same as for Algorithm 1, additionally: weight decay factor $\lambda$, momentum factor $\beta$,
   local momentum memory $\mathbf{v}_i^{(0)} := \mathbf{0} \; \forall i \in [n]$

---

Lines 1–8 in Algorithm 1 are left unmodified

Line 9 in Algorithm 1 is replaced with the following two lines

  9: $\mathbf{v}_i^{(t+1)} := (\mathbf{g}_i^{(t)} + \lambda \mathbf{x}_i^{(t)}) + \beta \mathbf{v}_i^{(t)}$        $\triangleleft$ local momentum with weight decay

  10: $\mathbf{x}_i^{(t+\frac{1}{2})} := \mathbf{x}_i^{(t)} - \eta \mathbf{v}_i^{(t+1)}$         $\triangleleft$ stochastic gradient update

---

**Setup.** In order to match the setting in (Tang et al., 2018) for our first set of experiments, we use a ring topology with $n = 8$ nodes and train the `ResNet20` architecture (He et al., 2016) on the `Cifar10` dataset (50K/10K training/test samples) (Krizhevsky, 2012). We randomly split the training data between workers and shuffle it after every epoch, following standard procedure as e.g. in (Goyal et al., 2017). We implement DCD and ECD with momentum (Tang et al., 2018), DeepSqueeze with momentum (Tang et al., 2019), CHOCO-SGD with momentum (Algorithm 2) and standard (all-reduce) mini-batch SGD with momentum and without compression (Dekel et al., 2012). Our implementations are open-source and available at `https://github.com/epfml/ChocoSGD`. The momentum factor is set to 0.9 without dampening. For all algorithms we fine-tune the initial learning rate and gradually warm it up from a relative small value (0.1) (Goyal et al., 2017) for the first 5 epochs. The learning rate is decayed by 10 twice, at 150 and 225 epochs, and stop training at 300 epochs. For CHOCO-SGD and DeepSqueeze the consensus learning rate $\gamma$ is also tuned. The detailed hyper-parameter tuning procedure refers to Appendix F. Every compression scheme is applied to every layer of `ResNet20` separately. We evaluate the top-1 test accuracy on every node separately over the whole dataset and report the average performance over all nodes.

**Compression Schemes.** We implement two *unbiased* compression schemes: (i) $\mathrm{gsgd}_b$ quantization that randomly rounds the weights to $b$-bit representations (Alistarh et al., 2017), and (ii) $\mathrm{random}_a$ sparsification, which preserves a randomly chosen $a$ fraction of the weights and sets the other ones to zero (Wangni et al., 2018). Further two *biased* compression schemes: (iii) $\mathrm{top}_a$, which selects the $a$ fraction of weights with the largest magnitude and sets the other ones to zero (Alistarh et al., 2018; Stich et al., 2018), and (iv) $\mathrm{sign}$ compression, which compresses each weight to its sign scaled by the norm of the full vector (Bernstein et al., 2018; Karimireddy et al., 2019). We refer to Appendix C for exact definitions of the schemes.

DCD and ECD have been analyzed only for unbiased quantization schemes, thus the combination with the two biased schemes is not supported by theory. In converse, CHOCO-SGD and DeepSqueeze has been studied only for biased schemes according to Definition 2. However, both unbiased compression schemes can be scaled down in order to meet the specification (cf. discussions in (Stich et al., 2018; Koloskova et al., 2019)) and we adopt this for the experiments.

**Results.** The results are summarized in Tab. 1. For unbiased compression schemes, ECD and DCD only achieve good performance when the compression ratio is small, and sometimes even diverge when the compression ratio is high. This is consistent[1] with the theoretical and experimental results in (Tang et al., 2018). We further observe that the performance of DCD with the biased $\mathrm{top}_a$ sparsification is much better than with the unbiased $\mathrm{random}_a$ counterpart, though this operator is not yet supported by theory.

CHOCO-SGD can generalize reasonably well in all scenarios (at most 1.65% accuracy drop) for fixed training budget. The $\mathrm{sign}$ compression achieves state-of-the-art accuracy and requires approximately $32\times$ less bits per weight than the full precision baseline.

---

[1] Tang et al. (2018) only consider absolute bounds on the quantization error. Such bounds might be restrictive (i.e. allowing only for low compression) when the input vectors are unbounded. This might be the reason for the instabilities observed here and also in (Tang et al., 2018, Fig. 4), (Koloskova et al., 2019, Figs. 5–6).

Table 1: Top-1 test accuracy for decentralized DCD, ECD, DeepSqueeze and CHOCO-SGD with different compression schemes. Reported top-1 test accuracies are averaged over three runs with fine-tuned hyper-parameters (learning rate, weight decay, consensus stepsize). The fine-tuned all-reduce baseline reaches accuracy 92.64, with 1.04 MB gradient transmission per iteration. ($\star$ indicates that 2 out of 3 runs diverged).

| Algorithm | Error-feedback | Quantization (QSGD) | | | | Sparsification (random-%) | | |
|---|---|---|---|---|---|---|---|---|
| | | 16 bits | 8 bits | 4 bits | 2 bits | 50% | 10% | 1% |
| transmitted data/iteration | | 0.52 MB | 0.26 MB | 0.13 MB | 0.065 MB | 1.04 MB | 0.21 MB | 0.031 MB |
| DCD-PSGD | ✗ | $92.51 \pm 0.05$ | $92.36 \pm 0.28$ | $23.56 \pm 2.97$ | diverges | $92.05 \pm 0.25$ | diverges | diverges |
| ECD-PSGD | ✗ | $92.02 \pm 0.14$ | $59.11 \pm 1.57$ | diverges | diverges | diverges | diverges | diverges |
| DeepSqueeze | ✓ | $92.27 \pm 0.21$ | $91.83 \pm 0.35$ | $91.47 \pm 0.21$ | $90.96 \pm 0.19$ | $91.46 \pm 0.09$ | $90.96 \pm 0.16$ | $88.55 \pm 0.11$ |
| CHOCO-SGD | ✓ | $92.34 \pm 0.19$ | $92.30 \pm 0.08$ | $91.92 \pm 0.27$ | $91.41 \pm 0.11$ | $92.54 \pm 0.26$ | $91.87 \pm 0.21$ | $91.32 \pm 0.17$ |

| Algorithm | Error-feedback | Sparsification (top-%) | | | Sign+Norm |
|---|---|---|---|---|---|
| | | 50% | 10% | 1% | - |
| transmitted data/iteration | | 1.04 MB | 0.21 MB | 0.031 MB | 0.032 MB |
| DCD-PSGD | ✗ | $92.40 \pm 0.11$ | $91.97 \pm 0.14$ | $89.79 \pm 0.40$ | $92.40 \pm 0.14$ |
| ECD-PSGD | ✗ | $17.03 \;\; \star$ | $16.78 \;\; \star$ | $18.03 \;\; \star$ | diverges |
| DeepSqueeze | ✓ | $91.55 \pm 0.28$ | $91.31 \pm 0.25$ | $90.47 \pm 0.17$ | $91.38 \pm 0.19$ |
| CHOCO-SGD | ✓ | $92.54 \pm 0.26$ | $92.29 \pm 0.05$ | $91.73 \pm 0.11$ | $92.46 \pm 0.10$ |

## 6 USE CASE I: ON-DEVICE PEER-TO-PEER LEARNING

We now shift our focus to challenging real-world scenarios which are intrinsically decentralized, i.e. each part of the training data remains local to each device, and thus centralized methods either fail or are inefficient to implement. Typical scenarios comprise e.g. sensor networks, or mobile devices or hospitals which jointly train a machine learning model. Common to these applications is that i) each device has only access to locally stored or acquired data, ii) communication bandwidth is limited (either physically, or artificially for e.g. metered connections), iii) the global network topology is typically unknown to a single device, and iv) the number of connected devices is typically large. Additionally, this fully decentralized setting is also strongly motivated by privacy aspects, enabling to keep the training data private on each device at all times.

**Modeling.** To simulate this scenario, we permanently split the training data between the nodes, i.e. the data is never shuffled between workers during training, and every node has distinct part of the dataset. To the best of our knowledge, no prior works studied this scenario for decentralized deep learning. For the centralized approach, gathering methods such as all-reduce are not efficiently implementable in this setting, hence we compare to the centralized baseline where all nodes route their updates to a central coordinator for aggregation. For the comparison we consider CHOCO-SGD with sign compression (this combination achieved the compromise between accuracy and compression level in Tab. 1)), decentralized SGD without compression (Lian et al., 2017), and centralized SGD without compression.

**Scaling to Large Number of Nodes.** To study the scaling properties of CHOCO-SGD, we train on $4, 16, 36$ and $64$ number of nodes. We compare decentralized algorithms on two different topologies: *ring* as the worst possible topology, and on the *torus* with much larger spectral gap. Their parameters are listed in the table 2. We train `ResNet8` (He et al., 2016) (78K parameters), on `Cifar10` dataset (50K/10K training/test samples) (Krizhevsky, 2012). For the simplicity, we keep the learning rate constant and separately tune it for all methods. We tune consensus learning rate for CHOCO-SGD.

Table 2: Summary of communication topologies.

| Topology | max. node degree | spectral gap $\rho$ | | | |
|---|---|---|---|---|---|
| | | $n = 4$ | $n = 16$ | $n = 36$ | $n = 64$ |
| ring | 2 | 0.67 | 0.05 | 0.01 | 0.003 |
| torus | 4 | 0.67 | 0.4 | 0.2 | 0.12 |
| fully-connected | $d$ | 1 | 1 | 1 | 1 |

The results are summarized in Fig. 1 (and Fig. 6, Tabs. 7–8 in Appendix G). First we compare the testing accuracy reached after 300 epochs (Fig. 1, *Left*). CentralizedSGD has a good performance for all the considered number of nodes. CHOCO-SGD slows down due to the influence of graph topology (`Decentralized` curve), which is consistent with the spectral gaps order (see Tab. 2), and also influenced by the communication compression (`CHOCO` curve), which slows down training uniformly for both topologies. We observed that the train performance is similar to the test on Fig. 1, therefore the performance degradation is explained by the slower convergence (Theorem 4.1) and is not a

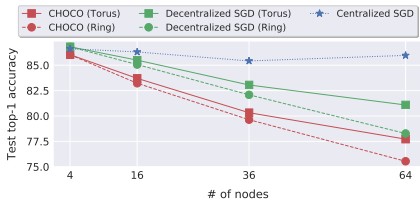 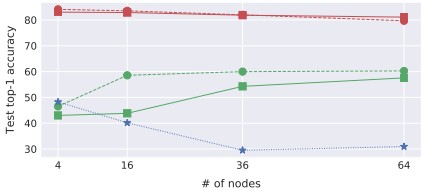

| Fix budget of 300 epochs | Fixed budget of communication size (1000 MB) |
| --- | --- |

Figure 1: Scaling of CHOCO-SGD with sign compression to large number of devices on Cifar10 dataset. *Left:* best testing accuracy of the algorithms reached after 300 epochs. *Right:* best testing accuracy reached after communicating 1000 MB.

generalization issue. Increasing the number of epochs improves the performance of the decentralized schemes. However, even using 10 times more epochs, we were not able to perfectly close the gap between centralized and decentralized algorithms for both train and test performance.

In the real decentralized scenario, the interest is not to minimize the epochs number, but the amount of communication to reduce the cost of the user's mobile data. We therefore fix the number of transmitted bits to 1000 MB and compare the best testing accuracy reached (Fig. 1, *Right*). CHOCO-SGD performs the best while having slight degradation due to increasing number of nodes. It is beneficial to use torus topology when the number of nodes is large because it has good mixing properties, for small networks there is not much difference between these two topologies—the benefit of large spectral gap is canceled by the increased communication due larger node degree for torus topology. Both Decentralized and Centralized SGD requires significantly larger number of bits to reach reasonable accuracy.

**Experiments on a Real Social Network Graph.** We simulate training models on user devices (e.g. mobile phones), connected by a real social network. We chosen Davis Southern women social network (Davis et al., 1941) with 32 nodes. We train ResNet20 (0.27 million parameters) model on the Cifar10 dataset (50K/10K training/test samples) (Krizhevsky, 2012) for image classification and a three-layer LSTM architecture (Hochreiter & Schmidhuber, 1997) (28.95 million parameters) for a language modeling task on WikiText-2 (600 training and 60 validation articles with a total of $2'088'628$ and $217'646$ tokens respectively) (Merity et al., 2016). The depicted curves of the training loss are the averaged local loss over all workers (local model with fixed local data); the test performance uses the mean of the evaluations for local models on whole test dataset. For more detailed experimental setup we refer to Appendix F.

The results are summarized in Figs. 2–3 and in Tab. 3. For the image classification task, when comparing the training accuracy reached after the same number of epochs, we observe that the decentralized algorithm performs best, follows by the centralized and lastly the quantized decentralized. However, the test accuracy is highest for the centralized scheme. When comparing the test accuracy reached for the same transmitted data[2], CHOCO-SGD significantly outperforms the exact decentralized scheme, with the centralized performing worst. We note a slight accuracy drop, i.e. after the same number of epochs (but much less transmitted data), CHOCO-SGD does not reach the same level of test accuracy than the baselines.

For the language modeling task, both decentralized schemes suffer a drop in the training loss when the evaluation reaching the epoch budget; while our CHOCO-SGD outperforms the centralized SGD in test perplexity. When considering perplexity for a fixed data volume (middle and right subfigure of Fig. 3), CHOCO-SGD performs best, followed by the exact decentralized and centralized algorithms.

On Figure 4 we additionally depict the test accuracy of the averaged model $\overline{\mathbf{x}}^{(t)} = \frac{1}{n} \sum_{i=1}^{n} \mathbf{x}_i^{(t)}$ (left) and averaged distance of the local models from the averaged model (right), for CHOCO-SGD on image classification task. Towards the end of the optimization the local models reach consensus (Figure 4, right), and their individual test performances are the same as performance of averaged model. Interestingly, before decreasing the stepsize at the epoch 225, the local models are in general

---

[2] The figure reports the transmitted data on the busiest node, i.e on the max-degree node (degree 14) node for decentralized schemes, and degree 32 for the centralized one.

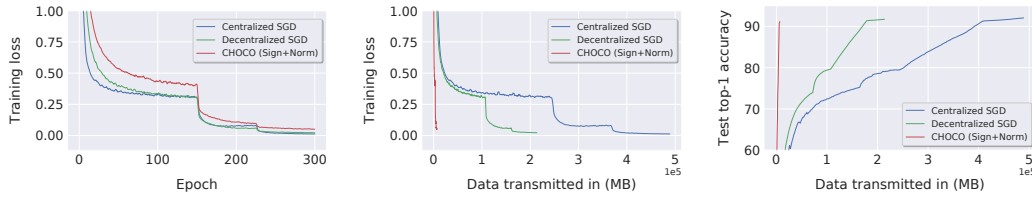

Figure 2: Image classification: ResNet-20 on CIFAR-10 on social network topology.

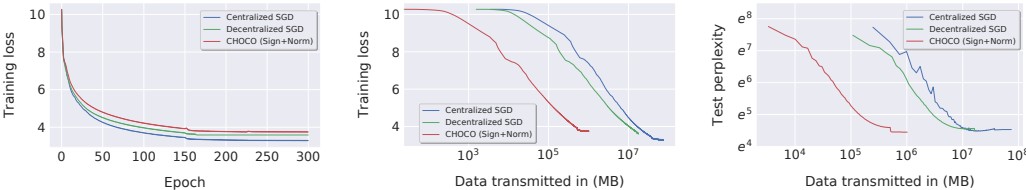

Figure 3: Language modeling: LSTM on WikiText-2 on social network topology.

Table 3: Summary of performance when training with the same epoch budget (as centralized SGD).

| Algorithm | max. connections/node | ResNet-20 (Fig. 2) | | LSTM (Fig. 3) | |
|---|---|---|---|---|---|
| | | data/gradient | top-1 test acc. | data/gradient | test perplexity |
| Centralized SGD | 32 | 1.04 MB | 93.00 | 110.43 MB | 89.39 |
| Exact Decentralized SGD | 14 | 1.04 MB | 92.12 | 110.43 MB | 91.38 |
| CHOCO-SGD (Sign + Norm) | 14 | 0.032 MB | 91.80 | 3.45 MB | 86.58 |

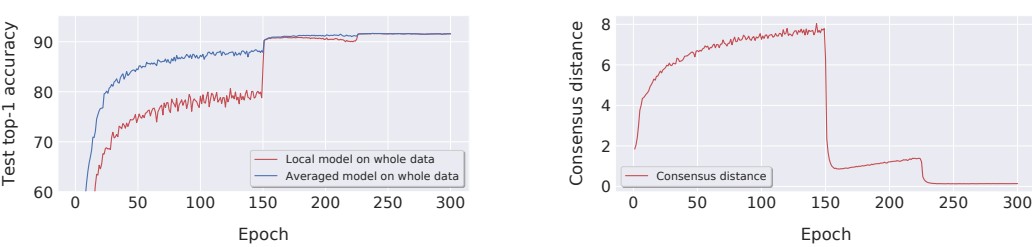

Figure 4: Parameter deviations for Resnet20 trained on Cifar10 (using CHOCO-SGD) on social network topology (32 workers). (Left) performance of the averaged model compared to the average of performances of local models. (Right) parameters divergence: averaged $L_2$ consensus distance between local models $\mathbf{x}_i$ and the averaged model $\bar{\mathbf{x}} = \frac{1}{n} \sum_{i=1}^{n} \mathbf{x}_i$, i.e., $\frac{1}{n} \sum_{i=1}^{n} \|\mathbf{x}_i - \bar{\mathbf{x}}\|_2^2$.

diverging from the averaged model, while decreasing only when the stepsize decreases. The same behavior was also reported in Assran et al. (2019).

## 7 USE CASE II: EFFICIENT LARGE-SCALE TRAINING IN A DATACENTER

Decentralized optimization methods offer a way to address scaling issues even for well connected devices, such as e.g. in datacenter with fast InfiniBand (100Gbps) or Ethernet (10Gbps) connections. Lian et al. (2017) describe scenarios when decentralized schemes can outperform centralized ones, and recently, Assran et al. (2019) presented impressive speedups for training on 256 GPUs, for the setting when all nodes can access all training data. The main differences of their algorithm to CHOCO-SGD are the asynchronous gossip updates, time-varying communication topology and most importantly exact communication, making their setup not directly comparable to ours. We note that these properties of asynchronous communication and changing topology for faster mixing are orthogonal to our contribution, and offer promise to be combined.

**Setup.** We train ImageNet-1k (1.28M/50K training/validation) (Deng et al., 2009) with Resnet-50 (He et al., 2016). We perform our experiments on 8 machines (n1-standard-32 from Google Cloud with Intel Ivy Bridge CPU platform), where each of machines has 4 Tesla P100

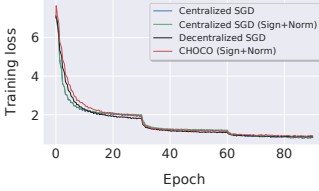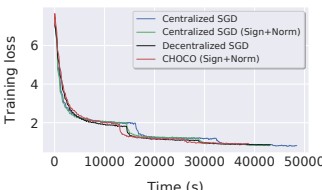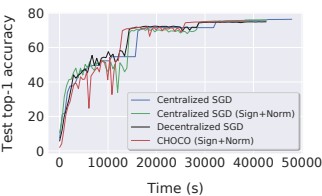

Figure 5: Large-scale training: `Resnet-50` on `ImageNet-1k` in the datacenter setting. The topology has 8 nodes (each accesses 4 GPUs). We use sign as the compression scheme, for CHOCO-SGD and Centralized SGD. For centralized SGD baseline without compression, we use all-reduce to aggregate the gradients; we use all-gather for centralized SGD with sign gradients quantization. The benefits of CHOCO-SGD can be further pronounced when scaling to more nodes.

GPUs and each machine interconnected via 10Gbps Ethernet. Within one machine communication is fast and we rely on the local data parallelism to aggregate the gradients for the later gradients communication (over the machines). Between different machines we consider centralized (fully connected topology) and decentralized (ring topology) communication, with and without compressed communication (sign compression). Several methods categorized by communication schemes are evaluated: (i) centralized SGD (full-precision communication), (ii) error-feedback centralized SGD with compressed communications Karimireddy et al. (2019) through sign compression, (iii) decentralized SGD (Lian et al., 2017) with parallelized forward pass and gradients communication (full-precision communication), and (iv) CHOCO-SGD with sign compressed communications. The mini-batch size on each GPU is 128, and we follow the general SGD training scheme in (Goyal et al., 2017) and directly use all their hyperparameters for all evaluated methods. Due to the limitation of the computational resource, we did not heavily tune the consensus stepsize for CHOCO-SGD[3].

**Results.** We depict the training loss and top-1 test accuracy in terms of epochs and time in Fig. 5. CHOCO-SGD benefits from its decentralized and parallel structure and takes less time than all-reduce to perform the same number of epochs, while having only a slight 1.5% accuracy loss[4]. In terms of time per epoch, our speedup does not match that of (Assran et al., 2019), as the used hardware and the communication pattern[5] are very different. Their scheme is orthogonal to our approach and could be integrated for better training efficiency. Nevertheless, we still demonstrate a time-wise 20% gain over the common all-reduce baseline, on our used commodity hardware cluster.

## 8 CONCLUSION

We propose the use of CHOCO-SGD (and its momentum version) for enabling decentralized deep learning training in bandwidth-constrained environments. We provide theoretical convergence guarantees for the non-convex setting and show that the algorithm enjoys linear speedup in the number of nodes. We empirically study the performance of the algorithm in a variety of settings on the image classification (ImageNet-1k, Cifar10) and on the language modeling task (WikiText-2). Whilst previous work successfully demonstrated that decentralized methods can be a competitive alternative to centralized training schemes when no communication constraints are present (Lian et al., 2017; Assran et al., 2019), our main contribution is to enable training in strongly communication-restricted environments, and while respecting the challenging constraint of locality of the training data. We theoretically and practically demonstrate the performance of decentralized schemes for arbitrary high communication compression, and under data-locality, and thus significantly expand the reach of potential applications of fully decentralized deep learning.

---

[3] We estimate the consensus stepsize by running CHOCO-SGD with different values for the first 3 epochs.

[4] Centralized SGD with full precision gradients achieved test accuracy of 76.37%, v.s. 76.03% for centralized SGD (with sign compression), v.s. 74.92% for plain decentralized SGD, and vs. 75.15% for CHOCO-SGD (with sign compression).

[5] We consider undirected communication, contrary to the directed 1-peer communication (every node sends and receives one message at every iteration) in Assran et al. (2019).

## ACKNOWLEDGEMENTS

We acknowledge funding from SNSF grant 200021_175796, as well as a Google Focused Research Award.

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

# A  CONVERGENCE OF CHOCO-SGD

In this section we present the proof of Theorem 4.1. For this, we will first derive a slightly more general statement: in Theorem A.3 we analyze CHOCO-SGD for arbitrary stepsizes $\eta$, and then derive Theorem 4.1 as a special case.

The structure of the proof follows Koloskova et al. (2019). That is, we first show that Algorithm 1 is a special case of a more general class of algorithms (given in Algorithm 3): Observe that Algorithm 1 consists of two main components: ② the stochastic gradient update, performed locally on each node, and ① the (quantized) averaging among the nodes. We can show convergence of all algorithms of this type—i.e. stochastic gradient updates ② followed by an arbitrary averaging step ①—as long as the averaging scheme exhibits linear convergence. For the specific averaging used in CHOCO-SGD, linear convergence has been shown in (Koloskova et al., 2019) and we will use their estimate of the convergence rate of the averaging scheme.

## A.1  A GENERAL FRAMEWORK FOR DECENTRALIZED SGD WITH ARBITRARY AVERAGING

For convenience, we use the following matrix notation in this subsection.

$$X^{(t)} := \left[\mathbf{x}_1^{(t)}, \ldots, \mathbf{x}_n^{(t)}\right] \in \mathbb{R}^{d \times n}, \qquad \overline{X}^{(t)} := \left[\overline{\mathbf{x}}^{(t)}, \ldots, \overline{\mathbf{x}}^{(t)}\right] \in \mathbb{R}^{d \times n},$$

$$\partial F(X^{(t)}, \xi^{(t)}) := \left[\nabla F_1(\mathbf{x}_1^{(t)}, \xi_1^{(t)}), \ldots, \nabla F_n(\mathbf{x}_n^{(t)}, \xi_n^{(t)})\right] \in \mathbb{R}^{d \times n}.$$

Decentralized SGD with arbitrary averaging is given in Algorithm 3.

---

**Algorithm 3** DECENTRALIZED SGD WITH ARBITRARY AVERAGING SCHEME

---

**input:** $X^{(0)} = \left[\mathbf{x}^{(0)}, \ldots, \mathbf{x}^{(0)}\right]$, stepsize $\eta$, averaging function $h : \mathbb{R}^{d \times n} \times \mathbb{R}^{d \times n} \to \mathbb{R}^{d \times n} \times \mathbb{R}^{d \times n}$,
   initialize $Y^{(0)} = 0$
  1: **for** $t$ **in** $0 \ldots T-1$ **do** {*in parallel for all workers* $i \in [n]$}
② { 2: $X^{(t+\frac{1}{2})} = X^{(t)} - \eta \partial F_i(X^{(t)}, \xi^{(t)})$     ◁ stochastic gradient updates
① { 3: $(X^{(t+1)}, Y^{(t+1)}) = h(X^{(t+\frac{1}{2})}, Y^{(t)})$    ◁ blackbox averaging/gossip
  4: **end for**

---

**Assumption 3.** *For an averaging scheme* $h \colon \mathbb{R}^{d \times n} \times \mathbb{R}^{d \times n} \to \mathbb{R}^{d \times n} \times \mathbb{R}^{d \times n}$ *let* $(X^+, Y^+) := h(X, Y)$ *for* $X, Y \in \mathbb{R}^{d \times n}$. *Assume that* $h$ *preserves the average of iterates:*

$$X^+ \frac{\mathbf{1}\mathbf{1}^\top}{n} = X \frac{\mathbf{1}\mathbf{1}^\top}{n}, \qquad\qquad \forall X, Y \in \mathbb{R}^{d \times n}, \qquad (4)$$

*and that it converges with linear rate for a parameter* $0 < c \leq 1$

$$\mathbb{E}_h \Psi(X^+, Y^+) \leq (1 - c)\Psi(X, Y), \qquad\qquad \forall X, Y \in \mathbb{R}^{d \times n}, \qquad (5)$$

*and Laypunov function* $\Psi(X, Y) := \|X - \overline{X}\|_F^2 + \|X - Y\|_F^2$ *with* $\overline{X} := \frac{1}{n} X \mathbf{1}\mathbf{1}^\top$, *where* $\mathbb{E}_h$ *denotes the expectation over internal randomness of averaging scheme* $h$.

**Example: Exact Averaging.** Setting $X^+ = XW$ and $Y^+ = X^+$ gives an exact consensus averaging algorithm with mixing matrix $W$ (Xiao & Boyd, 2004). It converges at the rate $c = \rho$, where $\rho$ is an eigengap of mixing matrix $W$, defined in Assumption 1. Substituting it into the Algorithm 3 we recover D-PSGD algorithm, analyzed in Lian et al. (2017).

**Example: CHOCO-SGD.** To recover CHOCO-SGD, we need to choose CHOCO-GOSSIP (Koloskova et al., 2019) as consensus averaging scheme, which is defined as $X^+ = X + \gamma Y(W - I)$ and $Y^+ = Y + Q(X^+ - Y)$ (in the main text we write $\hat{X}$ instead of $Y$). This scheme converges with $c = \frac{\rho^2 \delta}{82}$. The results from the main part can be recovered by

substituting this $c = \frac{\rho^2 \delta}{82}$ in the more general results below. It is important to note that for Algorithm 1 given in the main text, the order of the communication part ① and the gradient computation part ② is exchanged. We did this to better illustrate that both these parts are independent and that they can be executed in parallel. The effect of this change can be captured by changing the initial values but does not affect the convergence rate.

## A.2 PROOFS

**Remark A.1** (Mini-batch variance). *If for functions $f_i$, $F_i$ defined in (1) Assumption 2 holds, i.e. $\mathbb{E}_\xi \|\nabla F_i(\mathbf{x}, \xi) - \nabla f_i(\mathbf{x})\|^2 \leq \sigma_i^2, i \in [n]$, then*

$$\mathbb{E}_{\xi_1^{(t)}, \ldots, \xi_n^{(t)}} \left\| \frac{1}{n} \sum_{i=1}^n \left( \nabla f_i(\mathbf{x}_i^{(t)}) - \nabla F_i(\mathbf{x}_i^{(t)}, \xi_i^{(t)}) \right) \right\|^2 \leq \frac{\overline{\sigma}^2}{n}, \tag{6}$$

*where $\overline{\sigma}^2 = \frac{\sum_{i=1}^n \sigma_i^2}{n}$.*

*Proof.* This follows from

$$\mathbb{E} \left\| \frac{1}{n} \sum_{i=1}^n Y_i \right\|^2 = \frac{1}{n^2} \left( \sum_{i=1}^n \mathbb{E}\|Y_i\|^2 + \sum_{i \neq j} \mathbb{E}\langle Y_i, Y_j \rangle \right) = \frac{1}{n^2} \sum_{i=1}^n \mathbb{E}\|Y_i\|^2 \leq \frac{1}{n^2} \sum_{i=1}^n \sigma_i^2 = \frac{\overline{\sigma}^2}{n}$$

for $Y_i = f_i(\mathbf{x}_i^{(t)}) - \nabla F_i(\mathbf{x}_i^{(t)}, \xi_i^{(t)})$. Expectation of scalar product is equal to zero because $\xi_i$ is independent of $\xi_j$ since $i \neq j$. $\square$

**Lemma A.2.** *Under Assumptions 1–3 the iterates of the Algorithm 3 with constant stepsize $\eta$ satisfy*

$$\sum_{i=1}^n \left\| \overline{\mathbf{x}}^{(t)} - \mathbf{x}_i^{(t)} \right\|_2^2 \leq \eta^2 \frac{12nG^2}{c^2}.$$

*Proof of Lemma A.2.* We start by following the proof of Lemma 21 from Koloskova et al. (2019). Define $r_t = \mathbb{E} \left\| X^{(t)} - \overline{X}^{(t)} \right\|^2 + \mathbb{E} \left\| X^{(t)} - Y^{(t)} \right\|^2$,

$$
\begin{aligned}
r_{t+1} &\overset{(5)}{\leq} (1-c)\mathbb{E} \left\| \overline{X}^{(t+\frac{1}{2})} - X^{(t+\frac{1}{2})} \right\|_F^2 + (1-c)\mathbb{E} \left\| Y^{(t)} - X^{(t+\frac{1}{2})} \right\|_F^2 \\
&= (1-c)\mathbb{E} \left\| \overline{X}^{(t)} - X^{(t)} + \eta \partial F(X^{(t)}, \xi^{(t)}) \left( \frac{\mathbf{1}\mathbf{1}^\top}{n} - I \right) \right\|_F^2 \\
&\quad + (1-c)\mathbb{E} \left\| Y^{(t)} - X^{(t)} + \eta \partial F(X^{(t)}, \xi^{(t)}) \right\|_F^2 \\
&\overset{(9)}{\leq} (1-c)(1+\alpha^{-1})\mathbb{E} \left( \left\| \overline{X}^{(t)} - X^{(t)} \right\|_F^2 + \left\| Y^{(t)} - X^{(t)} \right\|_F^2 \right) \\
&\quad + (1-c)(1+\alpha)\eta^2 \mathbb{E} \left( \left\| \partial F(X^{(t)}, \xi^{(t)}) \left( \frac{\mathbf{1}\mathbf{1}^\top}{n} - I \right) \right\|_F^2 + \left\| \partial F(X^{(t)}, \xi^{(t)}) \right\|_F^2 \right) \\
&\leq (1-c) \left( (1+\alpha^{-1})\mathbb{E} \left( \left\| \overline{X}^{(t)} - X^{(t)} \right\|_F^2 + \left\| Y^{(t)} - X^{(t)} \right\|_F^2 \right) + 2n(1+\alpha)\eta^2 G^2 \right) \\
&\overset{\alpha = \frac{2}{c}}{\leq} \left( 1 - \frac{c}{2} \right) \mathbb{E} \left( \left\| \overline{X}^{(t)} - X^{(t)} \right\|_F^2 + \left\| Y^{(t)} - X^{(t)} \right\|_F^2 \right) + \frac{6n}{c}\eta^2 G^2.
\end{aligned}
$$

Define $A = 3nG^2$, we got a recursion

$$r_{t+1} \leq \left( 1 - \frac{c}{2} \right) r_t + \frac{2}{c}\eta^2 A,$$

Verifying that $r_t \leq \eta^2 \frac{4A}{c^2}$ satisfy recursion completes the proof as $\mathbb{E} \left\| X^{(t)} - \overline{X}^{(t)} \right\|^2 \leq r_t$.

Indeed, $r_0 = 0 \leq \eta^2 \frac{4A}{c^2}$ as $X^{(0)} = \overline{X}^{(0)}$ and $Y^{(0)} = 0$

$$r_{t+1} \leq \left(1 - \frac{c}{2}\right) r_t + \eta^2 \frac{2A}{c} \leq \left(1 - \frac{c}{2}\right) \eta^2 \frac{4A}{c^2} + \eta^2 \frac{2A}{c} = \eta^2 \frac{4A}{c^2}. \qquad \square$$

**Theorem A.3.** *Under Assumptions 1–3 with constant stepsize $\eta < \frac{1}{4L}$, the averaged iterates $\overline{\mathbf{x}}^{(t)} = \frac{1}{n} \sum_{i=1}^{n} \mathbf{x}_i^{(t)}$ of Algorithm 3 satisfy:*

$$\frac{1}{T+1} \sum_{t=0}^{T} \left\| \nabla f(\overline{\mathbf{x}}^{(t)}) \right\|_2^2 \leq \frac{4}{\eta(T+1)} \left( f(\overline{\mathbf{x}}^{(0)}) - f^\star \right) + \eta \frac{2\overline{\sigma}^2 L}{n} + \eta^2 \frac{36 G^2 L^2}{c^2}$$

*where $c$ denotes convergence rate of underlying averaging scheme.*

*Proof of Theorem A.3.* By $L$-smoothness

$$\mathbb{E}_{t+1} f(\overline{\mathbf{x}}^{(t+1)}) = \mathbb{E}_{t+1} f\left( \overline{\mathbf{x}}^{(t)} - \frac{\eta}{n} \sum_{i=1}^{n} \nabla F_i(\mathbf{x}_i^{(t)}, \xi_i^{(t)}) \right)$$

$$\leq f(\overline{\mathbf{x}}^{(t)}) \underbrace{- \mathbb{E}_{t+1} \left\langle \nabla f(\overline{\mathbf{x}}^{(t)}), \frac{\eta}{n} \sum_{i=1}^{n} \nabla F_i(\mathbf{x}_i^{(t)}, \xi_i^{(t)}) \right\rangle}_{=: T_1} + \mathbb{E}_{t+1} \underbrace{\frac{L}{2} \eta^2 \left\| \frac{1}{n} \sum_{i=1}^{n} \nabla F_i(\mathbf{x}_i^{(t)}, \xi_i^{(t)}) \right\|_2^2}_{=: T_2}$$

To estimate the second term, we add and subtract $\nabla f(\overline{\mathbf{x}}^{(t)})$

$$T_1 = -\eta \left\| \nabla f(\overline{\mathbf{x}}^{(t)}) \right\|^2 + \eta \left\langle \nabla f(\overline{\mathbf{x}}^{(t)}), \nabla f(\overline{\mathbf{x}}^{(t)}) - \frac{1}{n} \sum_{i=1}^{n} \nabla f_i(\mathbf{x}_i^{(t)}) \right\rangle$$

$$\overset{(8), \gamma = 1}{\leq} -\frac{\eta}{2} \left\| \nabla f(\overline{\mathbf{x}}^{(t)}) \right\|^2 + \frac{\eta}{2n} \sum_{i=1}^{n} \left\| \nabla f(\overline{\mathbf{x}}^{(t)}) - \nabla f_i(\mathbf{x}_i^{(t)}) \right\|^2$$

For the last term, we add and subtract $\nabla f(\overline{\mathbf{x}}^{(t)})$ and the sum of $\nabla f_i(\mathbf{x}_i^{(t)})$

$$T_2 = \mathbb{E}_{t+1} \left\| \frac{1}{n} \sum_{i=1}^{n} \left( \nabla F_i(\mathbf{x}_i^{(t)}, \xi_i^{(t)}) - \nabla f_i(\mathbf{x}_i^{(t)}) \right) \right\|_2^2 + \left\| \frac{1}{n} \sum_{i=1}^{n} \nabla f_i(\mathbf{x}_i^{(t)}) \pm \nabla f(\overline{\mathbf{x}}^{(t)}) \right\|_2^2$$

$$\overset{(6),(9),(7)}{\leq} \frac{\overline{\sigma}^2}{n} + \frac{2}{n} \sum_{i=1}^{n} \left\| f(\overline{\mathbf{x}}^{(t)}) - \nabla f_i(\mathbf{x}_i^{(t)}) \right\|_2^2 + 2 \left\| \nabla f(\overline{\mathbf{x}}^{(t)}) \right\|^2$$

Combining this together and using $L$-smoothness to estimate $\left\| f(\overline{\mathbf{x}}^{(t)}) - \nabla f_i(\mathbf{x}_i^{(t)}) \right\|_2^2$,

$$\mathbb{E}_{t+1} f(\overline{\mathbf{x}}^{(t+1)}) \leq f(\overline{\mathbf{x}}^{(t)}) - \eta \left( \frac{1}{2} - L\eta \right) \left\| \nabla f(\overline{\mathbf{x}}^{(t)}) \right\|_2^2 + \left( \frac{1}{2} \eta L^2 + \eta^2 L^3 \right) \frac{1}{n} \sum_{i=1}^{n} \left\| \overline{\mathbf{x}}^{(t)} - \mathbf{x}_i^{(t)} \right\|_2^2 + \frac{L\eta^2 \overline{\sigma}^2}{2n}.$$

Using Lemma A.2 to bound the third term and using that $\eta \leq \frac{1}{4L}$ in the second and in the third terms

$$\mathbb{E}_{t+1} f(\overline{\mathbf{x}}^{(t+1)}) \leq f(\overline{\mathbf{x}}^{(t)}) - \frac{\eta}{4} \left\| \nabla f(\overline{\mathbf{x}}^{(t)}) \right\|_2^2 + \eta^3 \frac{9 L^2 G^2}{c^2} + \eta^2 \frac{L\overline{\sigma}^2}{2n},$$

Rearranging terms and averaging over $t$

$$\frac{1}{T+1} \sum_{t=0}^{T} \left\| \nabla f(\overline{\mathbf{x}}^{(t)}) \right\|_2^2 \overset{(9)}{\leq} \frac{4}{\eta} \frac{1}{T+1} \sum_{t=0}^{T} \left( \mathbb{E} f(\overline{\mathbf{x}}^{(t)}) - \mathbb{E} f(\overline{\mathbf{x}}^{(t+1)}) \right) + \eta^2 \frac{36 G^2 L^2}{c^2} + \eta \frac{2 L \overline{\sigma}^2}{n}$$

$$\leq \frac{4}{\eta(T+1)} \left( f(\overline{\mathbf{x}}^{(0)}) - f^\star \right) + \eta \frac{2\overline{\sigma}^2 L}{n} + \eta^2 \frac{36 G^2 L^2}{c^2}$$

$$\square$$

**Corollary A.4.** *Under Assumptions 1–3 with constant stepsize* $\eta = \sqrt{\frac{n}{T+1}}$ *for* $T \geq 16nL^2$, *the averaged iterates* $\overline{\mathbf{x}}^{(t)} = \frac{1}{n}\sum_{i=1}^{n}\mathbf{x}_i^{(t)}$ *of Algorithm 3 satisfy:*

$$\frac{1}{T+1}\sum_{t=0}^{T}\left\|\nabla f(\overline{\mathbf{x}}^{(t)})\right\|_2^2 \leq \frac{4\left(f(\overline{\mathbf{x}}^{(0)}) - f^\star\right) + 2\overline{\sigma}^2 L}{\sqrt{n(T+1)}} + \frac{36G^2 nL^2}{(T+1)c^2}$$

*where* $c$ *denotes convergence rate of underlying averaging scheme.*

The first term shows a linear speed up compared to SGD on one node, whereas the underlying averaging scheme affects only the second-order term. Substituting the convergence rate for exact averaging with $W$ gives the rate $\mathcal{O}(1/\sqrt{nT} + n/(T\rho^2))$.

CHOCO-SGD with the underlying CHOCO-GOSSIP averaging scheme converges at the rate $\mathcal{O}(1/\sqrt{nT} + n/(T\rho^4\delta^2))$. The dependence on $\rho$ (eigengap of the mixing matrix $W$) is worse than in the exact case. This might either just be an artifact of our proof technique or a consequence of supporting arbitrary high compression.

## A.3 CONVERGENCE FOR ARBITRARY $T$

The previous result holds only for $T$ larger than $16nL^2$. This is not necessary and can be relaxed by carefully tuning the stepsize.

**Lemma A.5.** *For any parameters* $r_0 \geq 0, b \geq 0, e \geq 0, d \geq 0$ *there exists constant stepsize* $\eta \leq \frac{1}{d}$ *such that*

$$\Psi_T := \frac{r_0}{\eta(T+1)} + b\eta + e\eta^2 \leq 2\left(\frac{br_0}{T+1}\right)^{\frac{1}{2}} + 2e^{1/3}\left(\frac{r_0}{T+1}\right)^{\frac{2}{3}} + \frac{dr_0}{T+1}$$

*Proof.* Choosing $\eta = \min\left\{\left(\frac{r_0}{b(T+1)}\right)^{\frac{1}{2}}, \left(\frac{r_0}{e(T+1)}\right)^{\frac{1}{3}}, \frac{1}{d}\right\} \leq \frac{1}{d}$ we have three cases

- $\eta = \frac{1}{d}$ and is smaller than both $\left(\frac{r_0}{b(T+1)}\right)^{\frac{1}{2}}$ and $\left(\frac{r_0}{e(T+1)}\right)^{\frac{1}{3}}$, then

$$\Psi_T \leq \frac{dr_0}{T+1} + \frac{b}{d} + \frac{e}{d^2} \leq \left(\frac{br_0}{T+1}\right)^{\frac{1}{2}} + \frac{dr_0}{T+1} + e^{1/3}\left(\frac{r_0}{T+1}\right)^{\frac{2}{3}}$$

- $\eta = \left(\frac{r_0}{b(T+1)}\right)^{\frac{1}{2}} < \left(\frac{r_0}{e(T+1)}\right)^{\frac{1}{3}}$, then

$$\Psi_T \leq 2\left(\frac{r_0 b}{T+1}\right)^{\frac{1}{2}} + e\left(\frac{r_0}{b(T+1)}\right) \leq 2\left(\frac{r_0 b}{T+1}\right)^{\frac{1}{2}} + e^{\frac{1}{3}}\left(\frac{r_0}{(T+1)}\right)^{\frac{2}{3}},$$

- The last case, $\eta = \left(\frac{r_0}{e(T+1)}\right)^{\frac{1}{3}} < \left(\frac{r_0}{b(T+1)}\right)^{\frac{1}{2}}$

$$\Psi_T \leq 2e^{\frac{1}{3}}\left(\frac{r_0}{(T+1)}\right)^{\frac{2}{3}} + b\left(\frac{r_0}{e(T+1)}\right)^{\frac{1}{3}} \leq 2e^{\frac{1}{3}}\left(\frac{r_0}{(T+1)}\right)^{\frac{2}{3}} + \left(\frac{br_0}{T+1}\right)^{\frac{1}{2}}$$

$\square$

**Corollary A.6** (Generalized Theorem 4.1). *Under Assumptions 1–3 with constant stepsize* $\eta$ *tuned as in Lemma A.5, the averaged iterates* $\overline{\mathbf{x}}^{(t)} = \frac{1}{n}\sum_{i=1}^{n}\mathbf{x}_i^{(t)}$ *of Algorithm 3 satisfy:*

$$\frac{1}{T+1}\sum_{t=0}^{T}\left\|\nabla f(\overline{\mathbf{x}}^{(t)})\right\|_2^2 \leq 4\sqrt{\frac{2L\overline{\sigma}^2}{n(T+1)}} + 17\left(\frac{GLF_0}{c(T+1)}\right)^{\frac{2}{3}} + \frac{16LF_0}{T+1}$$

*where* $c$ *denotes convergence rate of underlying averaging scheme,* $F_0 = f(\overline{\mathbf{x}}^{(0)}) - f^\star$.

*Proof.* The result follows from Theorem A.3 and Lemma A.5 with $r_0 = 4\left(f(\overline{\mathbf{x}}^{(0)}) - f^\star\right)$, $b = \frac{2\bar{\sigma}^2 L}{n}$, $e = \frac{36G^2 L^2}{c^2}$ and $d = 4L$. $\qquad\square$

The corollary gives guarantees for the averaged vector of parameters $\overline{\mathbf{x}}$, however in a decentralized setting it is very expensive and sometimes impossible to average all the parameters distributed across several machines, especially when the number of machines and the model size is large. We can get similar guarantees on the individual iterates $\mathbf{x}_i$ as e.g. in (Assran et al., 2019). We summarize these briefly below.

**Corollary A.7** (Convergence of local weights). *Under the same setting as in Corollary A.6,*

$$\frac{1}{T+1} \sum_{t=0}^{T} \frac{1}{n} \sum_{i=1}^{n} \left\|\nabla f(\mathbf{x}_i^{(t)})\right\|_2^2 \leq 8\sqrt{\frac{2L\bar{\sigma}^2}{n(T+1)}} + 37\left(\frac{GLF_0}{c(T+1)}\right)^{\frac{2}{3}} + \frac{32LF_0}{T+1}$$

*Proof of Corollary A.7.*

$$\frac{1}{T+1} \sum_{t=0}^{T} \frac{1}{n} \sum_{i=1}^{n} \left\|\nabla f(\mathbf{x}_i^{(t)})\right\|_2^2 \leq \frac{1}{T+1} \sum_{t=0}^{T} \frac{1}{n} \sum_{i=1}^{n} \left(2\left\|\nabla f(\mathbf{x}_i^{(t)}) - \nabla f(\overline{\mathbf{x}}^{(t)})\right\|_2^2 + 2\left\|\nabla f(\overline{\mathbf{x}}^{(t)})\right\|_2^2\right)$$

$$\leq \frac{1}{T+1} \sum_{t=0}^{T} \frac{1}{n} \sum_{i=1}^{n} \left(2L^2 \left\|\mathbf{x}_i^{(t)} - \overline{\mathbf{x}}^{(t)}\right\|_2^2 + 2\left\|\nabla f(\overline{\mathbf{x}}^{(t)})\right\|_2^2\right)$$

where we used $L$-smoothness of $f$. Using Theorem A.3 and tuning the stepsize as in Lemma A.5 we get the statement of the corollary. $\qquad\square$

## B  USEFUL INEQUALITIES

**Lemma B.1.** *For arbitrary set of $n$ vectors $\{\mathbf{a}_i\}_{i=1}^n$, $\mathbf{a}_i \in \mathbb{R}^d$*

$$\left\|\sum_{i=1}^{n} \mathbf{a}_i\right\|^2 \leq n \sum_{i=1}^{n} \|\mathbf{a}_i\|^2 . \tag{7}$$

**Lemma B.2.** *For given two vectors $\mathbf{a}, \mathbf{b} \in \mathbb{R}^d$*

$$2\langle \mathbf{a}, \mathbf{b} \rangle \leq \gamma \|\mathbf{a}\|^2 + \gamma^{-1} \|\mathbf{b}\|^2 , \qquad\qquad \forall \gamma > 0 . \tag{8}$$

**Lemma B.3.** *For given two vectors $\mathbf{a}, \mathbf{b} \in \mathbb{R}^d$*

$$\|\mathbf{a} + \mathbf{b}\|^2 \leq (1 + \alpha) \|\mathbf{a}\|^2 + (1 + \alpha^{-1}) \|\mathbf{b}\|^2 , \qquad\qquad \forall \alpha > 0 . \tag{9}$$

*This inequality also holds for the sum of two matrices $A, B \in \mathbb{R}^{n \times d}$ in Frobenius norm.*

## C  COMPRESSION SCHEMES

We implement the compression schemes detailed below.

- $\mathrm{gsgd}_b$ (Alistarh et al., 2017). The unbiased $\mathrm{gsgd}_b \colon \mathbb{R}^d \to \mathbb{R}^d$ compression operator (for $b > 1$) is given as

$$\mathrm{gsgd}_b(\mathbf{x}) := \|\mathbf{x}\|_2 \cdot \mathrm{sig}(\mathbf{x}) \cdot 2^{-(b-1)} \cdot \left\lfloor \frac{2^{(b-1)} |\mathbf{x}|}{\|\mathbf{x}\|_2} + \mathbf{u} \right\rfloor$$

where $\mathbf{u} \sim_{u.a.r.} [0,1]^d$ is a random dithering vector and $\mathrm{sig}(\mathbf{x})$ assigns the element-wise sign: $(\mathrm{sig}(\mathbf{x}))_i = 1$ if $(\mathbf{x})_i \geq 0$ and $(\mathrm{sig}(\mathbf{x}))_i = -1$ if $(\mathbf{x})_i < 0$. As the value in the right bracket will be rounded to an integer in $\{0, \ldots, 2^{(b-1)} - 1\}$, each coordinate can be encoded with at most $(b-1) + 1$ bits (1 for the sign). For more efficent encoding schemes cf. Alistarh et al. (2017).

A biased version is given as

$$\text{gsgd}_b(\mathbf{x}) := \frac{\|\mathbf{x}\|_2}{\tau} \cdot \text{sig}(\mathbf{x}) \cdot 2^{-(b-1)} \cdot \left\lfloor \frac{2^{(b-1)} |\mathbf{x}|}{\|\mathbf{x}\|_2} + \mathbf{u} \right\rfloor$$

for $\tau = 1 + \min \left\{ \frac{d}{2^{2(b-1)}}, \frac{\sqrt{d}}{2^{(b-1)}} \right\}$ and is a $\delta = \frac{1}{\tau}$ compression operator (Koloskova et al., 2019).

- $\text{random}_a$ (Wangni et al., 2018). Let $\mathbf{u} \in \{0,1\}^d$ be a masking vector, sampled uniformly at random from the set $\{\mathbf{u} \in \{0,1\}^d : \|\mathbf{u}\|_1 = \lfloor ad \rfloor\}$. Then the unbiased $\text{random}_a : \mathbb{R}^d \to \mathbb{R}^d$ operator is defined as

$$\text{random}_a(\mathbf{x}) := \frac{d}{\lfloor ad \rfloor} \cdot \mathbf{x} \odot \mathbf{u}.$$

The biased version is given as

$$\text{random}_a(\mathbf{x}) := \mathbf{x} \odot \mathbf{u},$$

and is a $\delta = a$ compression operator (Stich et al., 2018).

Only $32\lfloor ad \rfloor$ bits are required to send $\text{random}_a(\mathbf{x})$ to another node—all the values of non-zero entries (we assume that entries are represented as float32 numbers). Receiver can recover positions of these entries if it knows the random seed of uniform sampling operator used to select these entries. This random seed could be communicated once on preprocessing stage (before starting the algorithm).

- $\text{top}_a$ (Alistarh et al., 2018; Stich et al., 2018). The biased $\text{top}_a : \mathbb{R}^d \to \mathbb{R}^d$ operator is defined as

$$\text{top}_a(\mathbf{x}) := \mathbf{x} \odot \mathbf{u}(\mathbf{x}),$$

where $\mathbf{u}(\mathbf{x}) \in \{0,1\}^d$, $\|\mathbf{u}\|_1 = \lfloor ad \rfloor$ is a masking vector with $(\mathbf{u})_i = 1$ for indices $i \in \pi^{-1}(\{1, \ldots, \lfloor ad \rfloor\})$ where the permutation $\pi$ is such that $|(\mathbf{x})_{\pi(1)}| \geq |(\mathbf{x})_{\pi(2)}| \geq \cdots \geq |(\mathbf{x})_{\pi(d)}|$. The $\text{top}_a$ operator is a $\delta = a$ compression operator (Stich et al., 2018).

In the case of $\text{top}_a$ compression $2 \cdot 32\lfloor ad \rfloor$ bits are required because along with the values we need to send positions of these values.

- sign (Bernstein et al., 2018; Karimireddy et al., 2019). The biased (scaled) $\text{sign} : \mathbb{R}^d \to \mathbb{R}$ compression operator is defined as

$$\text{sign}(\mathbf{x}) := \frac{\|\mathbf{x}\|_1}{d} \cdot \text{sgn}(\mathbf{x}).$$

The sign operator is a $\delta = \frac{\|\mathbf{x}\|_1^2}{d\|\mathbf{x}\|_2^2}$ compression operator (Karimireddy et al., 2019).

In total for the sign compression we need to send only $d + 32$ bits—one bit for every entry in $\mathbf{x}$ and 32 bits for $\|\mathbf{x}\|_1$.

## D    CHOCO-SGD WITH MOMENTUM

Algorithm 2 demonstrates how to combine CHOCO-SGD with weight decay and momentum. Nesterov momentum can be analogously adapted for our decentralized setting.

## E    ERROR FEEDBACK INTERPRETATION OF CHOCO-SGD

To better understand how does CHOCO-SGD work, we can interpret it as an error feedback algorithm (Stich et al., 2018; Karimireddy et al., 2019; Stich & Karimireddy, 2019). We can equivalently rewrite CHOCO-SGD (Algorithm 1) as Algorithm 4. The common feature of error feedback algorithms is that quantization errors are saved into the internal memory, which is added to the compressed value at the next iteration. In CHOCO-SGD the value we want to transmit is the difference $\mathbf{x}_i^{(t)} - \mathbf{x}_i^{(t-1)}$, which represents the evolution of local variable $\mathbf{x}_i$ at step $t$. Before compressing this value on line 4, the internal memory is added on line 3 to correct for the errors. Then, on line 5 internal memory is updated. Note that $\mathbf{m}_i^{(t)} = \mathbf{x}_i^{(t-1)} - \hat{\mathbf{x}}_i^{(t)}$ in the old notation.

---

**Algorithm 4** CHOCO-SGD (Koloskova et al., 2019) as Error Feedback

---

**input:** Initial values $\mathbf{x}_i^{(0)} \in \mathbb{R}^d$ on each node $i \in [n]$, consensus stepsize $\gamma$, SGD stepsize $\eta$,
     communication graph $G = ([n], E)$ and mixing matrix $W$, initialize $\hat{\mathbf{x}}_i^{(0)} = \mathbf{x}_i^{(-1)} := \mathbf{0}, \forall i \in [n]$

1: **for** $t$ **in** $0 \ldots T-1$ **do** {*in parallel for all workers* $i \in [n]$}
2:    $\mathbf{x}_i^{(t)} := \mathbf{x}_i^{(t-\frac{1}{2})} + \gamma \sum_{j:\{i,j\} \in E} w_{ij}\big(\hat{\mathbf{x}}_j^{(t)} - \hat{\mathbf{x}}_i^{(t)}\big)$         $\triangleleft$ modified gossip averaging
3:    $\mathbf{v}_i^{(t)} = \mathbf{x}_i^{(t)} - \mathbf{x}_i^{(t-1)} + \mathbf{m}_i^{(t)}$
4:    $\mathbf{q}_i^{(t)} := Q(\mathbf{v}_i^{(t)})$                     $\triangleleft$ compression
5:    $\mathbf{m}_i^{(t+1)} = \mathbf{v}_i^{(t)} - \mathbf{q}_i^{(t)}$                 $\triangleleft$ memory update
6:    **for** neighbors $j \colon \{i,j\} \in E$ (including $\{i\} \in E$) **do**
7:       Send $\mathbf{q}_i^{(t)}$ and receive $\mathbf{q}_j^{(t)}$               $\triangleleft$ communication
8:       $\hat{\mathbf{x}}_j^{(t+1)} := \mathbf{q}_j^{(t)} + \hat{\mathbf{x}}_j^{(t)}$                 $\triangleleft$ local update
9:    **end for**
10:    Sample $\xi_i^{(t)}$, compute gradient $\mathbf{g}_i^{(t)} := \nabla F_i(\mathbf{x}_i^{(t)}, \xi_i^{(t)})$
11:    $\mathbf{x}_i^{(t+\frac{1}{2})} := \mathbf{x}_i^{(t)} - \eta \mathbf{g}_i^{(t)}$                 $\triangleleft$ stochastic gradient update
12: **end for**

---

## F   DETAILED EXPERIMENTAL SETUP AND TUNED HYPERPARAMETERS

We precise the procedure of model training as well as the hyper-parameter tuning in this section.

**Social Network Setup.**   For the comparison we consider CHOCO-SGD with sign compression (this combination achieved the compromise between accuracy and compression level in Table 1)), decentralized SGD without compression, and centralized SGD without compression. We train two models, firstly ResNet20 (He et al., 2016) (0.27 million parameters) for image classification on the Cifar10 dataset (50K/10K training/test samples) (Krizhevsky, 2012) and secondly, a three-layer LSTM architecture (Hochreiter & Schmidhuber, 1997) (28.95 million parameters) for a language modeling task on WikiText-2 (600 training and 60 validation articles with a total of $2'088'628$ and $217'646$ tokens respectively) (Merity et al., 2016). For the language modeling task, we borrowed and adapted the general experimental setup of Merity et al. (2017), where we use a three-layer LSTM with hidden dimension of size 650. The loss is averaged over all examples and timesteps. The BPTT length is set to 30. We fine-tune the value of gradient clipping (0.4), and the dropout (0.4) is only applied on the output of LSTM.

We train both of ResNet20 and LSTM for 300 epochs, unless mentioned specifically. The per node mini-batch size is 32 for both datasets. The momentum (with factor 0.9) is only applied on the ResNet20 training.

**Social Network and a Datacenter details.**   For all algorithms, we gradually warmup (Goyal et al., 2017) the learning rate from a relative small value (0.1) to the fine-tuned initial learning rate for the first 5 training epochs. During the training procedure, the tuned initial learning rate is decayed by the factor of 10 when accessing $50\%$ and $75\%$ of the total training epochs. The learning rate is tuned by finding the optimal initial learning rate (after the scaling).

The optimal $\hat{\eta}$ is searched in a pre-defined grid and we ensure that the best performance was contained in the middle of the grids. For example, if the best performance was ever at one of the extremes of the grid, we would try new grid points. Same searching logic applies to the consensus stepsize.

Table 4 demonstrates the fine-tuned hpyerparameters of CHOCO-SGD for training ResNet-20 on Cifar10, while Table 6 reports our fine-tuned hpyerparameters of our baselines. Table 5 demonstrates the fine-tuned hpyerparameters of CHOCO-SGD for training ResNet-20/LSTM on a social network topology.

We estimate the runtime information (depicted in Figure 5) of different methods from three trials of the evaluation on Google Cloud (Kubernetes Engine). More precisely, we create the cluster on

Google Cloud for three times and each time we estimate the time per mini-batch of different methods (through the first two training epochs).

Table 4: Tuned hyper-parameters of CHOCO-SGD for training `ResNet-20` on `Cifar10`, corresponding to the ring topology with 8 nodes in Table 1. We randomly split the training data between nodes and shuffle it after every epoch. The per node mini-batch size is 128 and the degree of each node is 3.

| Compression schemes | Learning rate | Consensus stepsize |
|---|---|---|
| QSGD (16-bit) | 1.60 | 0.2 |
| QSGD (8-bit) | 0.96 | 0.2 |
| QSGD (4-bit) | 1.60 | 0.075 |
| QSGD (2-bit) | 0.96 | 0.025 |
| Sparsification (random-50%) | 2.40 | 0.45 |
| Sparsification (random-10%) | 1.20 | 0.075 |
| Sparsification (random-1%) | 0.48 | 0.00625 |
| Sparsification (top-50%) | 1.60 | 0.45 |
| Sparsification (top-10%) | 1.60 | 0.15 |
| Sparsification (top-1%) | 1.20 | 0.0375 |
| Sign+Norm | 1.60 | 0.45 |

Table 5: Tuned hyper-parameters of CHOCO-SGD, corresponding to the social network topology with 32 nodes in Table 3. We randomly split the training data between the nodes and keep this partition fixed during the entire training (no shuffling). The per node mini-batch size is 32 and the maximum degree of the node is 14.

| Configuration | Learning rate | Consensus stepsize |
|---|---|---|
| ResNet-20, Cifar10, Sign+Norm | 1.0 | 0.5 |
| LSTM, WikiText-2, Sign+Norm | 25 | 0.6 |

Table 6: Tuned hyper-parameters of DCD, ECD, and DeepSqueeze for training `ResNet-20` on `Cifar10`, corresponding to the ring topology with 8 nodes in Table 1. We randomly split the training data between nodes and shuffle it after every epoch. The per node mini-batch size is 128 and the degree of each node is 3. We only report the hpyerparameters corresponding to results that can reach to reasonable performance in our experiments.

| Compression schemes | Learning rate | Consensus stepsize |
|---|---|---|
| DCD, QSGD (16-bit) | 2.40 | - |
| DCD, QSGD (8-bit) | 1.20 | - |
| DCD, Sparsification (random-50%) | 0.80 | - |
| DCD, Sparsification (top-50%) | 1.20 | - |
| DCD, Sparsification (top-10%) | 1.60 | - |
| DCD, Sparsification (top-1%) | 2.40 | - |
| ECD, QSGD (16-bit) | 0.96 | - |
| ECD, QSGD (8-bit) | 1.20 | - |
| DeepSqueeze, QSGD (4-bit) | 0.60 | 0.01 |
| DeepSqueeze, QSGD (2-bit) | 0.80 | 0.005 |
| DeepSqueeze, Sparsification (top-50%) | 0.80 | 0.05 |
| DeepSqueeze, Sparsification (top-10%) | 0.60 | 0.01 |
| DeepSqueeze, Sparsification (top-1%) | 0.40 | 0.005 |
| DeepSqueeze, Sparsification (random-1%) | 0.80 | 0.0005 |
| DeepSqueeze, Sign+Norm | 0.48 | 0.01 |

# G ADDITIONAL PLOTS

To complement our results for scaling to a large number of nodes, we here additionally depict the learning curves (e.g. test accuracy) for the training on 64 nodes. We also mark the levels used for Fig. 1.

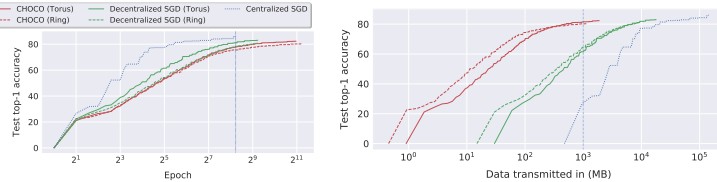

Figure 6: Scaling of CHOCO-SGD with sign compression to large number of devices on `Cifar10` dataset. Convergence curves for 64 nodes. Vertical lines corresponds to the epoch/bits budget used in Fig. 1.

Table 7: The exact epoch for the same bits budget in Fig. 1.

|  | $n = 4$ | $n = 16$ | $n = 36$ | $n = 64$ |
|---|---|---|---|---|
| Centralized | 5 | 6 | 6 | 6 |
| Decentralized (Ring) | 7 | 17 | 32 | 54 |
| Decentralized (Torus) | 6 | 10 | 18 | 29 |
| CHOCO (Ring) | 105 | 408 | 904 | 1588 |
| CHOCO (Torus) | 55 | 206 | 454 | 796 |

Table 8: The exact transmitted bits (in MB) for the same epoch budget in Fig. 1.

|  | $n = 4$ | $n = 16$ | $n = 36$ | $n = 64$ |
|---|---|---|---|---|
| Centralized | 139683 | 140041 | 144299 | 142899 |
| Decentralized (Ring) | 69841 | 17505 | 8016 | 4554 |
| Decentralized (Torus) | 139683 | 35010 | 16033 | 9109 |
| CHOCO (Ring) | 2208 | 564 | 253 | 144 |
| CHOCO (Torus) | 4417 | 1129 | 506 | 288 |

We additionally visualize the learning curves for the social network topology in Fig. 7 and Fig. 8.

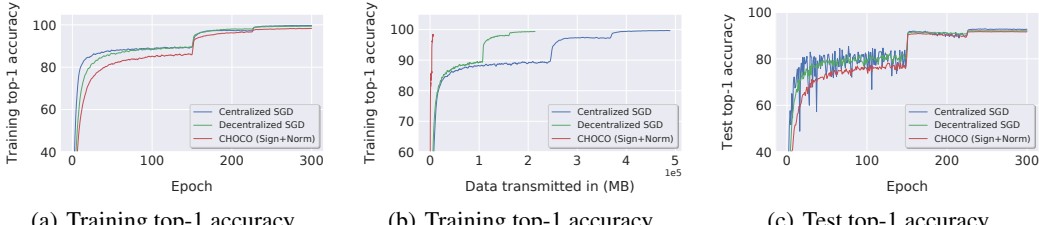

(a) Training top-1 accuracy.  (b) Training top-1 accuracy.  (c) Test top-1 accuracy.

Figure 7: Training `ResNet-20` on `CIFAR-10` with decentralized algorithm on a real world social network topology. The topology has 32 nodes and we assume each node can only access a disjoint subset of the whole dataset. The local mini-batch size is 32.

We additionally provide the learning curves of training top-1, top-5 accuracy and test top-5 accuracy for the datacenter experiment in Fig. 9.

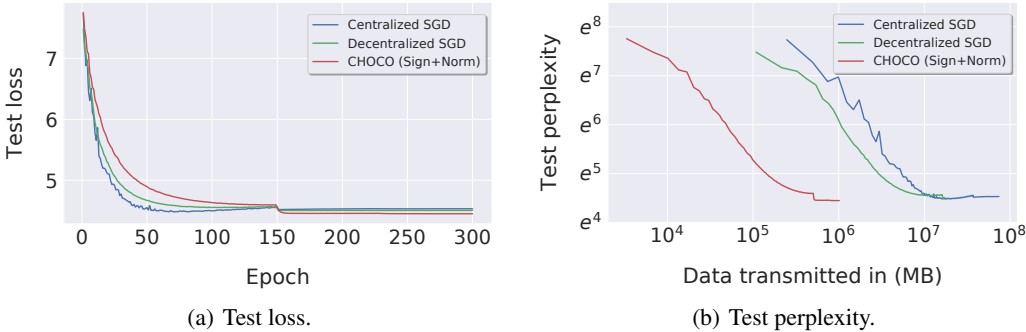

(a) Test loss.

(b) Test perplexity.

Figure 8: Training `LSTM` on `WikiText2` with decentralized algorithm on a real world social network topology. The topology has 32 nodes and we assume each node can only access a disjoint subset of the whole dataset. The local mini-batch size is 32.

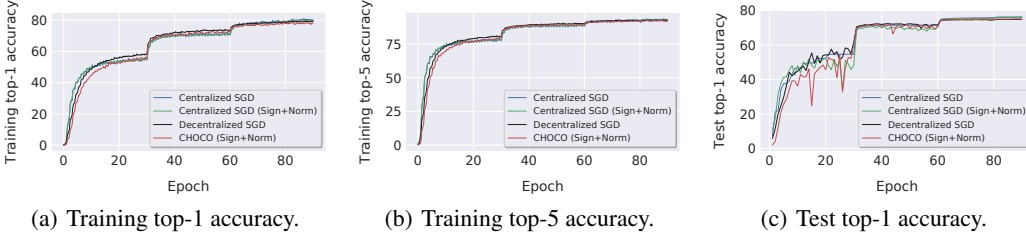

(a) Training top-1 accuracy.

(b) Training top-5 accuracy.

(c) Test top-1 accuracy.

Figure 9: Large-scale training: `ResNet-50` on `ImageNet` in the datacenter.

