# OpenReview forum: "Decentralized Deep Learning with Arbitrary Communication Compression"
_ICLR.cc/2020/Conference — Accept (Poster)_

### Official Review · AnonReviewer1 · 2019-10-23
**Official Blind Review #1**

**Rating:** 3

**Review:**

This paper studies the convergence of CHOCO-SGD for nonconvex objectives and shows its linear speedup while the original paper of CHOCO-SGD only provides analysis for convex objectives. The momemtum version of CHOCO-SGD is also provided although no theoretical analysis is presented.

Extensive empirical results are presented in this paper and the two use cases highlight some potential usage of the algorithm. However, there some concerns which could be addressed.

First, the authors only provide analysis on CHOCO-SGD but the comparison with baselines are based on their momemtum versions. Moreover, some highly relevant baseline like DeepSqueeze are not cited and compared. Thus, the advantage of vanilla CHOCO-SGD over other alternatives is not convincing.

Second, the cores of decentralized optimization include minimization of objective and consensus of the solution. However, no evaluation of the consensus is presented and this leads to the following point.

Third, it seems the authors report the average performance over all nodes using their individual model. If this is the case, the reported perfromance and comparison are not convincing. Without consensus, different nodes can have individual minimizer. In this case, the obtained average loss can be even smaller than the optimal loss. Under current measurement, if we run SGD on each worker individually without any communication, we will still get pretty good performance but this does not achieve the goal of decentralized optimization. Further clarification on this is needed.

Overall, I think the technical contribution of this paper is unclear and the evaluation is not convincing.


**Experience Assessment:**

I have published one or two papers in this area.

**Review Assessment: Checking Correctness Of Derivations And Theory:**

I assessed the sensibility of the derivations and theory.

**Review Assessment: Checking Correctness Of Experiments:**

I carefully checked the experiments.

**Review Assessment: Thoroughness In Paper Reading:**

I read the paper at least twice and used my best judgement in assessing the paper.

---

> ### Author Response · Authors · 2019-11-13
> **Response to Reviewer 1**
>
> Thank you for your valuable comments. We have addressed the main concerns that you indicated as the reason for your rating  ‘3: weak reject’ and we believe that we could sufficiently improve in all these aspects. Especially, we clarify below that our evaluation of the test performance is correct.
>
> [1. Comparison to DeepSqueeze]
> Thank you for pointing us to this highly related parallel work. In the revision we added DeepSqueeze to our comparison in Table 1 (all compression schemes on Cifar 10). The results for sign compression show that DeepSqueeze performs slightly worse than DCD and CHOCO-SGD. We will try to provide more of the missing values in the Table until the revision deadline (or latest for the final version).
>
> For the experiments we independently tuned the hyperparameters of DeepSqueeze, with the same grid search as for the other schemes (the grid in our search is dynamically extend, to make sure that the chosen values do not lie on the boundary and are indeed optimal), allowing for a fair comparison.
>
> [2. Consensus]
> We added evaluation of the consensus distance to the paper. (See Fig. 9).
>
> [3. Reporting of the performance]
> Please let us clarify: We mention on page 5 that “We evaluate the top-1 test accuracy on every node separately over the whole dataset and report the average performance over all nodes.” In formulas, this means that we report mean(g(x_i)), where g() measure the test accuracy (on the full test set) and x_i denotes the model on node i (this is the expected performance of a uniform random sampled model). We agree with you that if nodes would only evaluate performance with respect to a local part of the test set, then the results would not be convincing, but this is not what we report.
>
> Computing the averaged model \bar{x} = mean(x_i) requires one more full communication round (and might not be possible in the peer-to-peer decentralized setting).
>
> For completeness, we added the test performance g(\bar{x}) of the averaged model to the appendix (i.e. Figure 9 for the social network graph experiment with Resnet20 on Cifar10 dataset; we will run LSTM on WikiText-2 to obtain the corresponding plots as well).

---

### Official Review · AnonReviewer2 · 2019-10-23
**Official Blind Review #2**

**Rating:** 6

**Review:**

This paper studies non-convex decentralized optimization with arbitrary communication compression. It is well motivated and well written. The authors consider CHOCO-SGD for non-convex decentralized optimization and establish the convergence result based on the compression ratio. This result does not rely on specific quantized method and main term in the upper bound matches with the centralized baseline. The authors also show CHOCO-SGD with momentum is effectiveness in practical. The experimental results on several benchmark datasets validate the algorithm achieves better performance than baselines

Both of the theoretical and empirical results are convincing. I believe this paper is ready for publication.

Minor comments:

1. It is prefer to present Algorithm 2 and 3 in the main text, since they are mentioned by the statement of Theorem 4.1 and used in experiments respectively.

2. Can you provide some theoretical guarantee of CHOCO-SGD with momentum?


**Experience Assessment:**

I have read many papers in this area.

**Review Assessment: Checking Correctness Of Derivations And Theory:**

I assessed the sensibility of the derivations and theory.

**Review Assessment: Checking Correctness Of Experiments:**

I assessed the sensibility of the experiments.

**Review Assessment: Thoroughness In Paper Reading:**

I read the paper at least twice and used my best judgement in assessing the paper.

---

> ### Author Response · Authors · 2019-11-13
> **Response to Reviewer 2**
>
> Thank you for your positive assessment of our work. We have moved the pseudo code of the proposed momentum version of CHOCO-SGD to the main text as you suggested. Thus, we would like to ask you if you could reconsider your score to align it with your very positive comments (‘I believe this paper is ready for publication.’)
>
> 1. We decided to keep the pseudo code of CHOCO-SGD with general averaging (Algorithm 3 in the revision) in the appendix, to avoid to introduce additional notation in the main text and to keep the presentation of the main results as clean as possible.
>
> 2. We agree that it would be nice to have theoretical guarantees for the momentum version of Choco-SGD, however, this was not a focus in this paper. We are not aware of a result in the literature that can (theoretically) prove a strict advantage of SGD with momentum over vanilla SGD. We think that answers in that much simpler setting should be derived first, before attempting the proof of the decentralized momentum scheme.

---

### Official Review · AnonReviewer3 · 2019-10-23
**Official Blind Review #3**

**Rating:** 6

**Review:**

The authors present an algorithm CHOCO-SGD to make use of communication compression in a decentralized setting. This is an interesting problem and the results are promising. Firstly they prove the convergence rate of the algorithm on non-convex smooth functions, which shows a nearly linear speedup.

Second, on the practical part, there have 3 main results:
	1. They compare CHOCO-SGD under various compression schemes with the baseline. The results show the algorithm generally outperforms the baseline.
	2. They implement it over a realistic peer-to-peer social network and show a great communication performance under such a network with limited bandwidth.
	3. In a datacenter setting, they compare the algorithm with all-reduce, which is a centralized communication method. The results show a strong training reduction for CHOCO-SGD.

Also, the paper is mostly nicely written.

However, there have several issues:

	1. In the introduction, they introduce their experiments with the order from "datacenter experiment" to "peer-to-peer experiment", which is different from the actual presenting order.
	2. In the description of Algorithm 1, the representation of initial values should be x{(-1/2)}_{i} instead of x{(0)}_{i} since line 2 using the term x^{t-1/2}_{i} with the range of t from 0 to T-1.
	3. About "datacenter setting" experiment, it seems not an apple to apple comparison between CHOCO-SGD and all-reduce method since CHOCO-SGD stands for the decentralized algorithm with compression and all-reduce stands for a centralized algorithm without compression. It's better to compare with at least one centralized algorithm with a compression scheme (like QSGD[1], signSGD[2], DGC[3]).
	4. Although they compare with the baseline (DCD and ECD) on Cifar-10 dataset,  it's worth to compare with them on the ImageNet since the result may be different under large-scale training.

Overall, this could be a great paper if fixing the issues above.


[1] D. Alistarh, D. Grubic, J. Z. Li, R. Tomioka, and M. Vojnovic. QSGD: Communication-efﬁcient SGD via gradient quantization and encoding. In Proc. Advances in Neural Information Processing Systems (NIPS), 2017.

[2] Bernstein J, Zhao J, Azizzadenesheli K, Anandkumar A. signSGD with majority vote is communication efficient and fault tolerant. arXiv. 2018 Oct 11.

[3] Lin Y, Han S, Mao H, Wang Y, Dally WJ. Deep gradient compression: Reducing the communication bandwidth for distributed training. arXiv preprint arXiv:1712.01887. 2017 Dec 5.

**Experience Assessment:**

I have published one or two papers in this area.

**Review Assessment: Checking Correctness Of Derivations And Theory:**

I assessed the sensibility of the derivations and theory.

**Review Assessment: Checking Correctness Of Experiments:**

I assessed the sensibility of the experiments.

**Review Assessment: Thoroughness In Paper Reading:**

I read the paper at least twice and used my best judgement in assessing the paper.

---

> ### Author Response · Authors · 2019-11-13
> **Response to Reviewer 3**
>
> Thank you for your positive assessment of our work.
>
> 1.&2. Thank you for spotting this. We addressed these comments in the revision.
>
> [Experiments on Imagenet]
> From Table 1 we can deduce that ECD has difficulties to converge even on the smaller Resnet-20 architecture. Similarly, DCD does consistently perform worse that CHOCO-SGD thus we believe that we will see similar differences on the large scale Imagenet training.
>
> To allow for better comparison, we will add one of your suggested centralized baselines to these plots.

---

### Author Response · Authors · 2019-11-13
**General Comments on Revision 1**

We would like to thank the reviewers for their useful comments. We have fixed all typos and included a few additional numerical experiments that where requested (some experiments are still running/scheduled and we plan to update our draft again on Friday with additional results).

---

> ### Author Response · Authors · 2019-11-15
> **Revision 2**
>
> We thank the reviewers again for useful comments.
>
> We did our best to provide new results for the additional experiments.

---

### Decision · Program_Chairs · 2019-12-19

**Decision:**

Accept (Poster)

**Comment:**

The authors present an algorithm CHOCO-SGD to make use of communication compression in a decentralized setting. This is an interesting problem, and the paper is well-motivated and well-written. On the theoretical side, the authors prove the convergence rate of the algorithm on non-convex smooth functions, which shows a nearly linear speedup. The experimental results on several benchmark datasets validate the algorithm achieves better performance than baselines. These can be made more convincing by comparing with more baselines (including DeepSqueeze and other centralized algorithms with a compression scheme), and on larger datasets. The authors should also clarify results on consensus.